# AdaSG: A Lightweight Feature Point Matching Method Using Adaptive Descriptor with GNN for VSLAM

**DOI:** 10.3390/s22165992

**Published:** 2022-08-11

**Authors:** Ye Liu, Kun Huang, Jingyuan Li, Xiangting Li, Zeng Zeng, Liang Chang, Jun Zhou

**Affiliations:** 1School of Information and Communication Engineering, University of Electronic Science and Technology of China, Chengdu 611731, China; 2School of Microelectronics, Shanghai University, Shanghai 200444, China

**Keywords:** feature point matching, GNN, VSLAM

## Abstract

Feature point matching is a key component in visual simultaneous localization and mapping (VSLAM). Recently, the neural network has been employed in the feature point matching to improve matching performance. Among the state-of-the-art feature point matching methods, the SuperGlue is one of the top methods and ranked the first in the CVPR 2020 workshop on image matching. However, this method utilizes graph neural network (GNN), resulting in large computational complexity, which makes it unsuitable for resource-constrained devices, such as robots and mobile phones. In this work, we propose a lightweight feature point matching method based on the SuperGlue (named as AdaSG). Compared to the SuperGlue, the AdaSG adaptively adjusts its operating architecture according to the similarity of input image pair to reduce the computational complexity while achieving high matching performance. The proposed method has been evaluated through the commonly used datasets, including indoor and outdoor environments. Compared with several state-of-the-art feature point matching methods, the proposed method achieves significantly less runtime (up to 43× for indoor and up to 6× for outdoor) with similar or better matching performance. It is suitable for feature point matching in resource constrained devices.

## 1. Introduction

Feature point matching [1,2] is a key component for pose estimation in VSLAM, which is widely used in applications, such as autonomous driving, robots, and wearable augmented reality (AR) [3,4,5,6]. Pose estimation based on feature points mainly consists of two parts: feature point extraction and feature point matching. The former extracts the feature points from two stereo or consecutive images and represents them using certain descriptors [7,8,9]. The latter computes the distance of the feature points based on the descriptors [10] and performs feature point matching to estimate the pose (as shown in Figure 1). Most of the conventional feature point matching methods use the nearest neighbor method [11] to compute the distance between the feature points and perform the matching.

One of the major challenges for the feature point matching is that it is prone to errors due to numerous factors, such as viewpoint change, lack of texture, illumination variation, and blur. To address this challenge, various methods have been proposed, such as Lowe’s ratio test [12], mutual checks, and neighborhood consensus [13,14], which combines the nearest neighbor with outlier rejection methods.

In recent years, neural networks [15,16] have been utilized to enhance the quality of the descriptors by learning from a large number of datasets to improve matching performance. One of the most state-of-the-art methods is the SuperGlue [17], which is inspired by Transformer [18], and utilizes the GNN with attention mechanism to improve the matching performance. It ranked the first in the CVPR 2020 workshop on image matching [19]. Despite its superior performance, the SuperGlue brings in large computational complexity, making it unsuitable for resource-constrained devices, such as robots and mobile phones.

There are some lightweight methods proposed to improve the heavy-weight algorithm. Adjusting the neural network, which is possible for many neural networks, is usually used to reduce parameters [20,21,22]. Moreover, taking advantage of the characteristics of some specific application scene can always reduce computational complexity [23,24].

Inspired by these methods, in this work, a lightweight feature point matching method (named AdaSG) based on SuperGlue has been proposed in order to address the above issue. It adaptively adjusts the descriptor and the outlier rejection mechanism according to the similarity of the input image pair so as to reduce the computational complexity while maintaining good matching performance.

The remaining part of this paper is organized as follows: Section 2 reviews the existing works on feature point matching. Section 3 presents the proposed AdaSG method. Section 4 discusses the experimental results, and Section 5 concludes the paper.

## 2. Related Work

The feature point matching method mainly consists of two steps: distance calculation and outlier rejection [12]. The former is responsible for calculating the distance between different feature points based on their descriptors by using certain distance metrics, such as Euclidean distance [25], inner product [26], or Hamming distance [27]. Based on the distance, the outlier rejection filters out the feature points with large distances, which are defined as outliers. The most common outlier rejection method is the nearest neighbor, which sets a threshold to select the closest feature points. However, this method does not work well in certain scenarios, such as viewpoint change, lack of texture, illumination variation, and blur. To address these challenges, the revised outlier rejection methods have been proposed. For example, Lowe’s ratio test was proposed in [12]. In this work, the outlier rejection takes the distance ratio of the closest neighbor and the second closest neighbor into account. The correct matchings are determined by comparing this ratio with a certain threshold. In [13], the neighborhood consensus was proposed, in which the invariant regions are extracted to facilitate the matching when the viewpoint changes. However, in some challenging scenarios with significant viewpoint changes, it is still difficult to obtain correct matching using these methods.

To deal with this challenge, some works started using neural networks to enhance the descriptors of feature points to obtain better matching results [28,29,30]. These studies mainly use the convolutional neural networks (CNN) to obtain better descriptors than the handcrafted descriptors from feature point extraction. In [31], LF-Net, which is composed of two stages networks, was proposed to extract feature points location and descriptors. The former detects feature points and the latter computes descriptors. In [32], an encoder–decoder architecture was trained by using homographic adaption to learn multi-scale features. This work not only enhances the descriptors but also boosts feature point detection. In [33], D2-Net was proposed to extract the feature points distribution and their descriptors by using a shared CNN. This describe-and-detect strategy performs well in the aforementioned challenging scenarios. In [34], R2D2 was proposed to jointly learn reliable detectors and descriptors. This work discovered that descriptors learned in discriminative regions help improve the matching performance.

Other studies leverage regional information to improve the discrimination of descriptors, such as regional descriptors [35] or log-polar patches [36]. In [35], the generation of descriptors utilizes not only visual context from high-level image representation but also geometric context from 2D feature point distribution. In [36], the local region information is extracted with a log-polar sampling scheme to acquire better descriptors, which oversamples the adjacent regions of the feature points and undersamples the regions far away from it.

Although the neural network-based methods have improved the matching performance to some extent, they have not achieved satisfactory results. In these works, there is an issue limiting their performance. These works focus on pixel-level information but lack higher-level information which reflects the correspondences between the feature points. In fact, feature point matching can be seen as an instance of graph matching. A graph can be constructed by associating each feature point to a node and defining special edges. How to leverage the knowledge in the graph domain has become an interesting topic.

Recently, GNN [17,37,38,39] has been utilized to address this issue. In [37], the PointNet was proposed to learn the information of both global and local feature points for 3D recognition tasks. PointNet architecture is a special case of GNN but without definition of edges. Inspired by [37], PointCN [38] was proposed to deal with feature point matching by classifying inliers and outliers. In this work, a novel normalization technique is proposed to process each feature point separately with embedded global information. The global information is generated by taking the camera motion into account. However, PointNet-like architecture cannot capture the local context information. Based on [38], OANet [39] was proposed to capture the local context information by exploiting differentiable pooling which is used to process an entire graph. In the meanwhile, the global context information is also captured by an Order-Aware Filtering block with spatial association. Based on these works, the SuperGlue was recently proposed, which applies an attentional GNN to aggregate information based on self-cross attention mechanism [17]. This method learned complex priors and elegantly solved occlusion and non-repeatable feature points. The SuperGlue ranked first in the CVPR 2020 workshop on image matching [19].

The methods based on neural network outperform conventional methods by enhancing the descriptors. However, this significantly increases the computational complexity, making them unsuitable for resource-constrained devices, such as robots and mobile phones. Therefore, how to reduce the computational complexity while maintaining good matching performance needs to be investigated.

Some work has proposed lightweight methods to reduce computational complexity. Exploring more lightweight neural network is usually used to reduce parameters. In [20], MobileNet is proposed to build a lightweight deep neural network. Moreover, leveraging the characteristics of certain specific scenarios can always reduce computational complexity. [23] uses a small, annotated dataset to build a prediction model, which avoids using complex neural networks to complete tasks. In [24], ORBSLAM2 uses a constant velocity movement model to reduce matching complexity during tracking threads.

## 3. Proposed AdaSG

We have chosen the SuperGlue as the baseline of our proposed method due to its excellent performance. The SuperGlue consists of two major blocks: an attentional GNN and an optimal matching layer, as shown in Figure 2. The attentional GNN carries out two strategies to obtain better descriptors. One is to integrate more contextual cues (e.g., the position of feature points) to increase the distinctiveness of original descriptors. The other is to use alternating self- and cross-attention layers to resolve ambiguities. This strategy, inspired by humans, integrates more knowledge between both images during matching. When asked to match feature points in two images, humans usually look back-and-forth between the two images until finding a difference or similarity.

After that, the optimal matching layer performs iterative optimization to obtain the matching results, which is represented as an assignment matrix. The assignment matrix is obtained by maximizing the score function as in normal optimization problems through iterative procedure. A dustbin mechanism is used to assign unmatched feature points to reduce incorrect matching. This optimization problem can be regarded as an optimal transport problem. Therefore, the solution is a Sinkhorn algorithm, which is a commonly used efficient algorithm for optimal transport problem.

Despite its excellent matching performance, the SuperGlue brings in large computational complexity in two aspects. Firstly, the SuperGlue treats all the input images equally. In order to achieve good matching performance, the images are always processed using GNN regardless of the similarity of the input image pair. However, not all images are difficult to match and require a complex GNN to enhance the quality of the descriptors. For the input images that are easy to match, a simplified method can significantly reduce the computational complexity while achieving a similar performance to the SuperGlue. 

Secondly, the optimal matching layer brings in additional computational complexity due to the iteration of Sinkhorn algorithm. The runtime of Sinkhorn with iteration almost accounts for half of the overall runtime. How to reduce the complexity of the optimal matching layer is worth investigating.

To address the above issue, in this work, we propose Adaptive SuperGlue (AdaSG in short), which is a lightweight feature point matching method. In this method, the descriptor and the outlier rejection mechanism are adaptively adjusted according to the similarity of the input image pairs so as to reduce the computational complexity, while achieving good matching performance.

AdaSG is mainly composed of three modules: a similarity evaluation module, an adaptive aggregation module and an adaptive outlier rejection module, as shown in Figure 3.

### 3.1. Similarity Evaluation Module

The similarity evaluation module is responsible for evaluating the similarity of the input image pair. If the similarity of the two images is low (e.g., as shown in Figure 4), meaning that a large viewpoint change or illumination change is present, then the matching difficulty is considered high. In this case, the AdaSG will turn on the D-Mode. In this mode, the adaptive aggregation module will activate the GNN for the descriptor enhancement, and the outlier rejection module will increase the optimization strength to improve the matching performance. 

If the similarity of the two images is high (e.g., as shown in Figure 5), meaning that the viewpoint change or illumination change is small, then the matching difficulty is considered low. In this case, the AdaSG will turn on the E-Mode. In this mode, the aggregation module will de-activate the GNN to skip the descriptor enhancement and the outlier rejection module will lower the optimization strength to reduce the computational complexity.

The similarity of the two images is calculated using the sum of absolute differences (SAD), as shown in (1).
(1)$$Dscore=\sum_{i=1}^{W}\sum_{j=1}^{H}\left|I_{A}(i,j)-I_{B}(i,j)\right|$$
where *Dscore* represents the similarity and $I_{A}\left(i,j\right)$ and $I_{B}\left(i,j\right)$ represent the pixel grayscale value of image A and image B at (i,j).

### 3.2. Adaptive Aggregation Module

The adaptive aggregation module is used to enhance the input descriptors by activating the GNN to improve the matching performance according to the work modes (i.e., D-mode and E-mode). The architecture of the adaptive aggregation module is shown in Figure 6 and its detailed operation is described below.

The inputs of the adaptive aggregation module include the descriptors and the location information of the feature points from the image pair. A feature point encoder (FPE) is used to fuse the descriptors and the location information according to (2).
(2)xi(0)=di+MLP (pi)
where xi(0) represents the new descriptor; $d_{i}$ represents the original descriptor; $p_{i}$ represents the location information of the feature point, which is composed of the coordinates and confidence of the feature point; and Multilayer Perceptron (*MLP*) represents a neural network composed of fully connected layers.

After the FPE, a *GNN* is used to enhance the descriptor of feature points by aggregating the information of the descriptors based on the graph structure with special edges defined by attention mechanism as in the SuperGlue [17]. The architecture of the *GNN* is shown in Figure 7. The *GNN* is mainly composed of a self-layer and cross-layer alternatively, which repeats nine times. The architecture of the self-layer and cross-layer are the same, which leverages the attention mechanism to obtain an aggregation message. However, their inputs are different. The inputs of the self-layers are from one of the input images while the inputs of the cross-layer are from both input images. Through the self-cross mechanism, both self-image and cross-image information are aggregated to improve the discriminativeness and matching performance.

Different from [17], which uses *GNN* to enhance the descriptors for all the input images, in the proposed adaptive aggregation method, the descriptor is adaptively generated according to the work mode, which is determined by the similarity evaluation module from the input images. For the D-mode, the *GNN* and FPE are activated to enhance the descriptors. In this case, the descriptors are the output of the *GNN*, as shown in (3).
(3)fiA=GNN(xA(0),xB(0)),∀i∈AfjB=GNN(xB(0),xA(0)),∀j∈B
where xA(0) and xB(0) indicate the output of FPE for the input image pair (*A*, *B*), respectively, and $f_{i}{}^{A}$ and $f_{j}{}^{B}$ are descriptors generated by the *GNN* from the image A and B.

For the E-mode, the *GNN* and FPE will be de-activated for skipping the descriptor enhancement. In this case, the descriptors are the original descriptors, which are good enough for matching as the two images are very similar, as shown in (4).
(4)$$\begin{array}{c} f_{i}{}^{A}=d_{i}\;\;,\forall i\in A\\ f_{j}{}^{B}=d_{j}\;\;,\forall j\in B \end{array}$$
where $f_{i}{}^{A}$ and $f_{j}{}^{B}$ are the descriptors of image *A* and image *B*. The (3) and (4) can be merged and represented using (5).
(5)fiA=γ∗GNN(xA(0),xB(0))+(1−γ)∗di,∀i∈AfjB=γ∗GNN(xB(0),xA(0))+(1−γ)∗di,∀j∈B
where γ∈{0,1}, γ=0 indicates that E-mode is selected, while γ=1 indicates that D-mode is selected.

### 3.3. Outlier Rejection Module

In the SuperGlue, the outlier rejection module employs Sinkhorn and mutual-check to filter out incorrect matching based on the distance matrix. Different from the SuperGlue, in this work, we proposed adaptive Sinkhorn iterations to reduce complexity while maintaining the performance. Through experiments we found that in the E-mode, the Sinkhorn, with a few iterations (e.g., 3), can achieve high accuracy, while in the D-mode, many more iterations (e.g., 100) are needed for achieving high accuracy. Therefore, the number of iterations can also be adapted according to the work mode. In the E-mode, the Sinkhorn iteration is set to a smaller number, while it is set to a larger number in the D-mode. Additionally, the mutual-check mechanism, which is used to calculate the distance of feature points and select similar feature points, can be adapted according to the work mode.

In the D-mode, the original descriptor vector has been transformed by the *GNN*, therefore the inner product is used to compute the distance matrix S, as shown in (6). The *i*th feature point in image A and the *j*th feature point in image B can be matched if $S_{i,j}$ is the maximum both in the *i*th row and *j*th column of S and larger than the point selection threshold.
(6)$$S_{i,j}=<f_{i}^{A},f_{j}^{B}>,\forall \left(i,j\right)\in A\times B$$
where < > represents the inner product. $f_{i}^{A}$ represents the *i*th descriptor of image A and $f_{j}^{B}$ represents the *j*th descriptor of image B. $S_{i,j}$ represents the element in the S at (i,j).

In the E-mode, without GNN the Euclidean distance can be used to compute the distance of original descriptor vectors, which is also shown in (7). The *i*th feature point in image A and the *j*th feature point in image B can be matched if $S_{i,j}$ is the minimum both in *i*th row and *j*th column of S and less than the point selection threshold.
(7)$$S_{i,j}=\sqrt{{\left(f_{i}^{A}-f_{j}^{B}\right)}^{T}\left(f_{i}^{A}-f_{j}^{B}\right)},\forall \left(i,j\right)\in A\times B$$

## 4. Experimental Results and Analysis

### 4.1. Experimental Setup

The proposed AdaSG is implemented using PyTorch and run on an NVIDIA Tesla v100 GPU. As in the SuperGlue paper [17], two datasets are used to evaluate the performance of the proposed method, including the indoor dataset Scannet [40] and the outdoor dataset YFCC100M [41]. The Scannet is a large-scale indoor dataset composed of sequence images with ground truth poses. YFCC100M is a large-scale outdoor dataset composed of non-sequence images with ground truth poses. These two datasets can be used to evaluate the performance of the matching method for indoor and outdoor scenarios. However, YFCC100M does not contain sequence images. Therefore, we also used KITTI [42], which is also a large-scale outdoor dataset with sequence images. Both YFCC100M and KITTI have been used to evaluate the proposed AdaSG for outdoor scenarios. For the Scannet and the YFCC100M datasets, all the sequences tested in the SuperGlue paper [17] are used to evaluate the performance. For the KITTI dataset, all the images are used for evaluation to investigate the performance on sequence images.

Like the previous work [17,38,39], the area under the cumulative error curve (AUC) of the pose error at different threshold is used to evaluate the performance of the proposed method. The pose error is the maximum of angle error between rotation and translation. The cumulative error curve is generated by calculating the precision. Then AUC@m indicates the area under this curve up to a maximum threshold m. In this experiment, m is set to 5°, 10°, and 20°.

The matching precision (P) and the matching score (MS) are also presented to measure the matching performance based on its epipolar distance. The matching is correct if its epipolar distance is less then $10^{-4}$. The matching precision is the percentage of the correct matched feature point pairs in all the matched feature point pairs, and the matching score is the percentage of the correct matched feature point pairs in all the feature point pairs, including matched and unmatched pairs.

As mentioned previously, in the proposed AdaSG method, three threshold parameters need to be set, including the threshold of similarity of the input image pair, $T_{S}$; the point selection threshold in D-mode, $T_{D}$; and the point selection threshold in E-mode, $T_{E}$. For the experiments, these parameters are set to 0.12, 0.2, and 0.8, respectively. In the E-mode the Sinkhorn iteration $T_{i}$ is set to 0, and in the D-mode the Sinkhorn iteration $T_{i}$ is set to 100.

The parameters of SuperGlue and AdaSG are shown in Table 1. SuperGlue contains a Keypoint Encoder, which has 100 k parameters and an attentional GNN with 12 M parameters. Compared to SuperGlue, AdaSG has zero parameters in E-mode and has the same parameters as SuperGlue in D-mode.

For benchmarking, the proposed AdaSG is compared with the state-of-the-art matching methods, including the SuperGlue [17], the PointCN [38], the OANet [39], and the GlusterGNN [43]. The OANet and the PointCN are both combined with the nearest neighbor method for testing. The GlusterGNN is only compared on the YFCC100M, which is the same as in [43], for fair comparison. Firstly, as the compared methods use different datasets and evaluation metrics in their papers, for fair comparison, we obtained the source codes of the compared methods and re-ran them using the same datasets and evaluation metrics, as in [17]. Secondly, as the compared methods are matching methods, for the feature point extraction before the matching we used the SuperPoint, which is a state-of-the-art feature point extraction method [32] for all the compared methods. In addition, for fair comparison with the SuperGlue, the weights of the GNN in the proposed AdaSG use the same weights from the SuperGlue paper [17].

Furthermore, to investigate and compare the performance of the AdaSG and the SuperGlue on resource-constrained devices, we implemented the AdaSG and the SuperGlue on an embedded system board (i.e., RK3399Pro [44]), which has four ARM-A53 cores at 1.4 GHz with 4 Gb DDR3 for running the algorithms, as shown in Figure 8. In the experiments, the images are sent from the laptop to the board for feature point extraction and matching, and the matching results are sent to the monitor for display.

### 4.2. Indoor Dataset Experimental Results

Indoor matching is challenging because there are many walls, floors, and other areas where the texture information is not rich. Additionally, there are many objects with high similarity, such as doors and rooms, which can easily cause mismatch. Figure 9 and Table 2 show indoor dataset experimental results using the Scannet dataset. As can be seen in the table, the performance of the proposed AdaSG is similar to that of the SuperGlue and is better than the other methods. However, its average runtime is significantly shorter than that of the SuperGlue (43×). This is mainly due to the fact that AdaSG adaptively adjusts its architecture according to the similarity of the input image pair which significantly reduces the computational complexity.

We also investigated the impact of threshold of similarity on the matching precision and average runtime by varying the threshold value. The results are shown in Figure 10. It can be seen from the figure that as the threshold value increases, the matching precision decreases and the runtime decreases. When the threshold increases from 0 to 0.12, which is the selected threshold value, the average runtime decreases by 97.67%, while the matching precision only decreases slightly from 96.62% to 96.48%. The average runtime and matching precision saturate at 0.001 s and 96.48%, respectively, when the threshold exceeds 0.1, because with such large threshold all the input image pairs are consider similar and the E-mode is activated.

### 4.3. Outdoor Dataset Experimental Results

Figure 11 and Table 3 show the outdoor dataset experimental results using YFCC100M. As can be seen from the table, the performance of the AdaSG is similar to that of the SuperGlue and is significantly better than the other methods. However, its average runtime is also similar to that of the SuperGlue, which is longer than the other methods. This is mainly due to the fact that the YFCC100M dataset mainly contains non-sequence images and in this case D-mode is activated much more frequently to improve the matching performance at the cost of runtime.

Figure 12 and Table 4 show the outdoor dataset experimental results using KITTI. The KITTI contains sequence images, but the moving speed is quite high as the images are taken in a moving car. Therefore, it is a mixture of similar and non-similar image pairs. This causes the D-mode and E-mode to be activated in a hybrid fashion. As can be seen in the table, the performance of the AdaSG is similar to that of SuperGlue, which is much better than the other methods, and its average runtime is almost 6× better than the SuperGlue. Figure 13 and Figure 14 show the impact of the threshold of similarity on the matching precision and average runtime for the YFCC100M and KITTI datasets, respectively. As the YFCC100M dataset mainly contains non-sequence images (with low similarity), the average run time only decreases by 0.94% when the threshold increases from 0 to 0.12. The situation becomes different for the KITTI dataset which contains sequence images. The average run time decreases by 83.04% with a small decrease on the matching precision (from 99.79% to 99.76%) when the threshold increases from 0 to 0.12.

### 4.4. Experimental Results on Embedded System

Figure 15 and Figure 16 show the experimental results on the embedded system RK3399Pro for the KITTI dataset. As can be seen from the Figure 16, the matching performance of the AdaSG is similar to that of the SuperGlue on different sequences. Figure 15 shows the runtime comparison of the AdaSG and the SuperGlue. It can be seen that the average runtime of AdaSG is around 10× less than SuperGlue on the sequences 01, 03, 04, 06, 09, and 10, and is around 3× less than SuperGlue on the sequence 00, 02, 05, 07, and 08. Compared with the SuperGlue, the proposed AdaSG is much faster when running on resource-constrained devices.

## 5. Conclusions

The SuperGlue is one of the top feature point matching methods (ranked the first in the CVPR 2020 workshop on image matching). This method uses GNN to improve the matching performance. However, this also brings in large computational complexity making it unsuitable for resource-constrained devices. In this work, we propose a lightweight feature point matching method based on the SuperGlue (named AdaSG). It adaptively adjusts its operating architecture according to the similarity of the input image pair to reduce the computational complexity while achieving high matching performance. When running on the GPU, the proposed method achieves up to a 43× and 6× average runtime reduction for the indoor and outdoor datasets, respectively, with similar or higher matching performance compared with the SuperGlue. When running on an embedded system with constrained computing resources, the proposed method achieves up to 10× performance improvement compared with the SuperGlue.

## Figures and Tables

**Figure 1 sensors-22-05992-f001:**
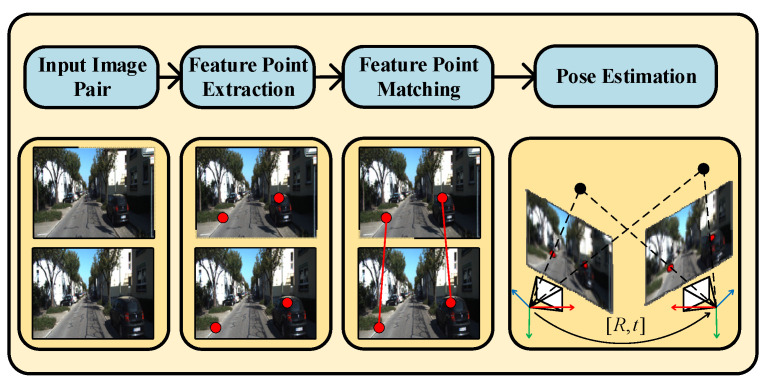
The operating flow of pose estimation in VSLAM. The red points represent the feature points in the images and the black points represent the corresponding 3D points in the real world.

**Figure 2 sensors-22-05992-f002:**
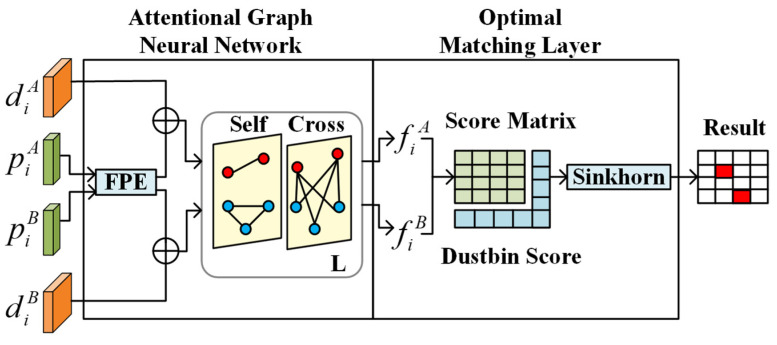
The architecture of SuperGlue. SuperGlue uses a keypoint encoder, which is also called a feature point encoder (FPE), to fuse contextual cues (keypoint position p and descriptor d) and then uses alternating self- and cross-attention layers (repeated L times) to obtain matching descriptors f.

**Figure 3 sensors-22-05992-f003:**
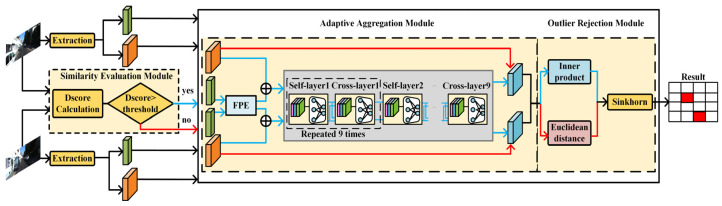
The architecture of the proposed AdaSG. Red lines represent the E-mode dataflow and blue lines represent the D-mode dataflow. Self-layering and cross-layering were repeated nine times alternatively.

**Figure 4 sensors-22-05992-f004:**
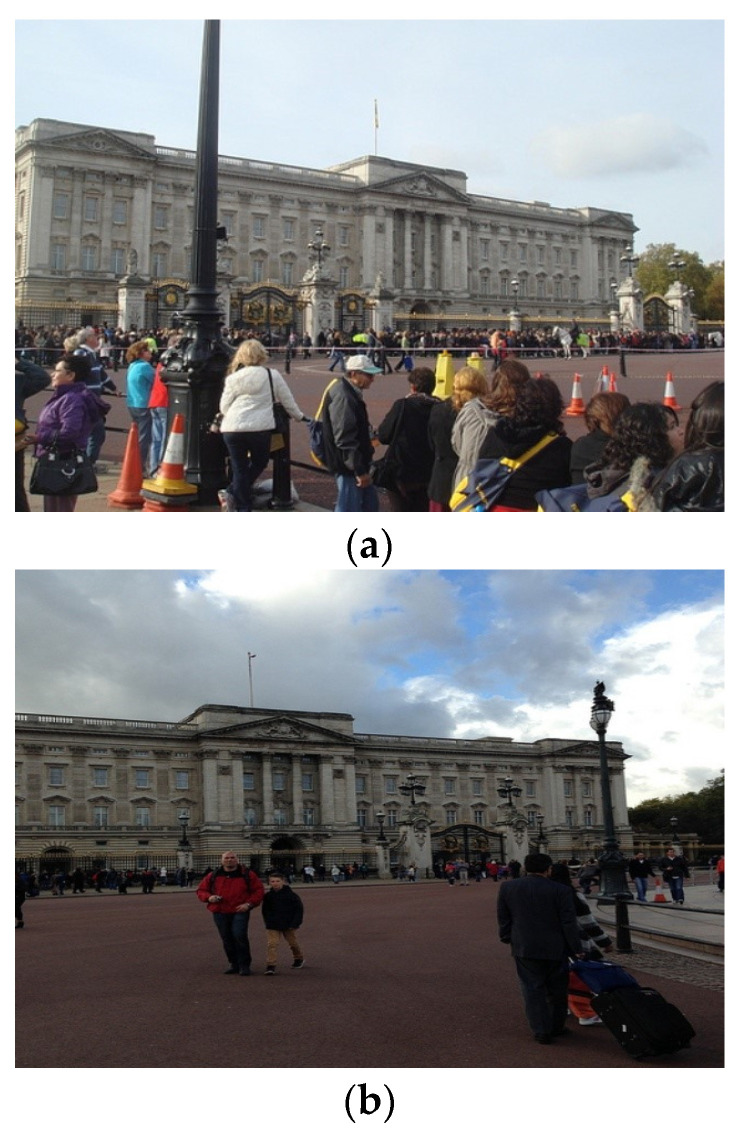
The image pair with low similarity. (**a**,**b**) are pictures of a landmark location, but the camera angle and image content are very different.

**Figure 5 sensors-22-05992-f005:**
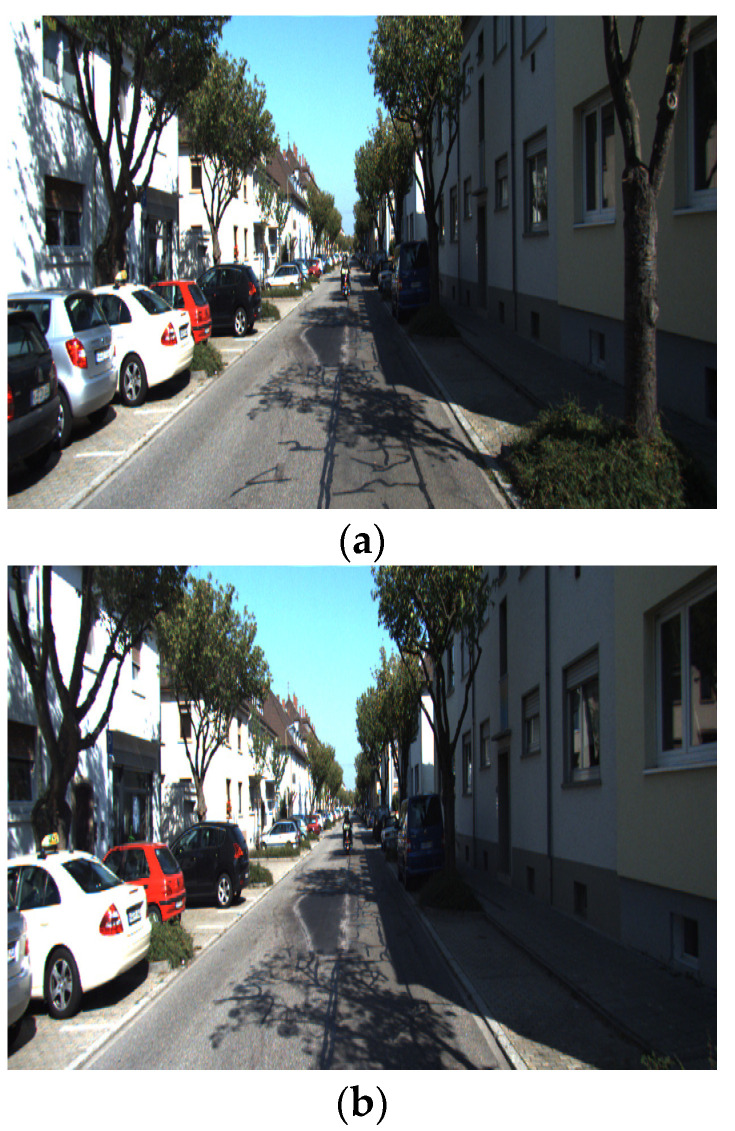
The image pair with high similarity. (**a**,**b**) are two images extracted from the video stream, which look almost identical.

**Figure 6 sensors-22-05992-f006:**
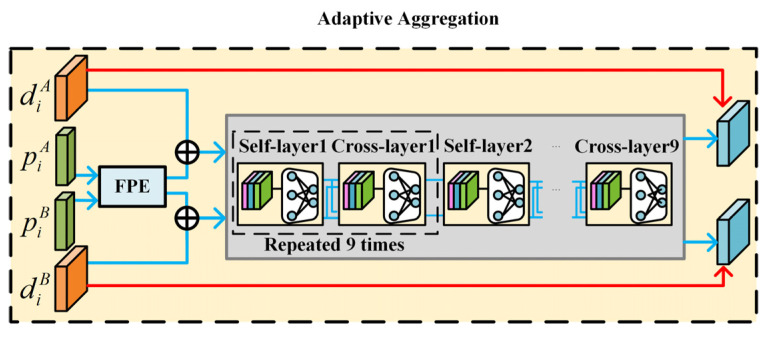
The architecture of adaptive aggregation module. The red lines represent E-mode dataflow and the blue lines represent D-mode dataflow. Self-layering and cross-layering are repeated nine times alternatively.

**Figure 7 sensors-22-05992-f007:**
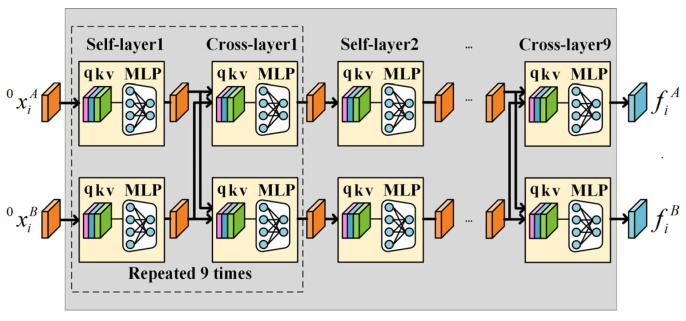
The architecture of GNN. The q, k, and v represent the query, key, and value in the attention mechanism. x is obtained by FPE. The self-layer and cross-layer are repeated nine times alternatively.

**Figure 8 sensors-22-05992-f008:**
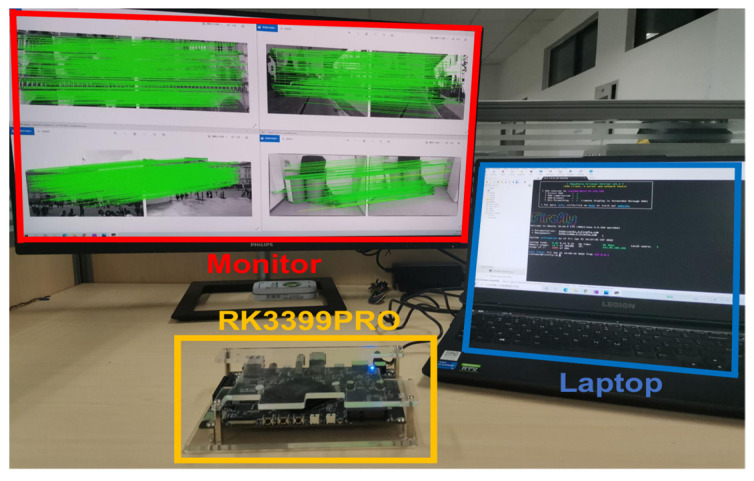
The whole system.

**Figure 9 sensors-22-05992-f009:**
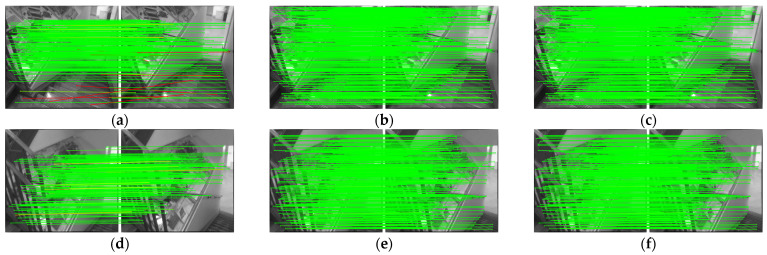
The visualized matching result on Scannet. The red lines represent mismatches, and the green lines represent correct matching. (**a**) The results of OANet. (**b**) The results of SuperGlue. (**c**) The results of AdaSG. (**d**) The other results of OANet. (**e**) The other results of SuperGlue. (**f**) The other results of AdaSG.

**Figure 10 sensors-22-05992-f010:**
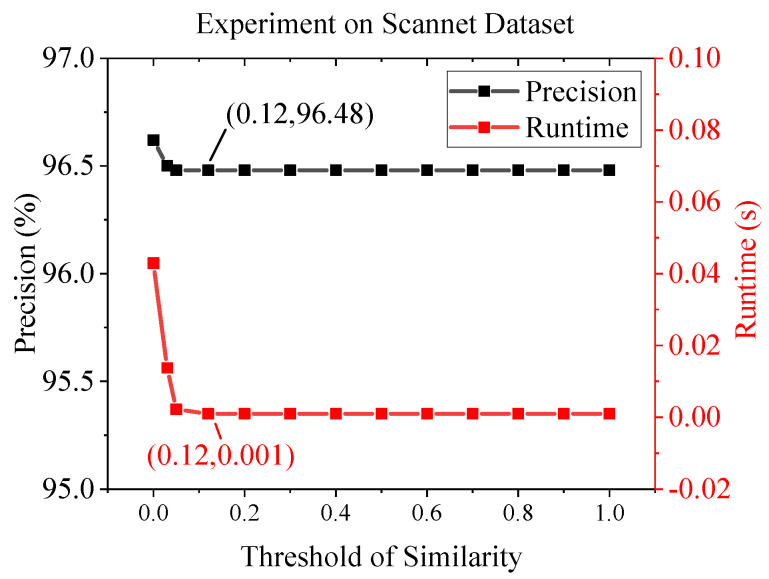
The impact of threshold of similarity on matching precision and average runtime on the Scannet dataset.

**Figure 11 sensors-22-05992-f011:**
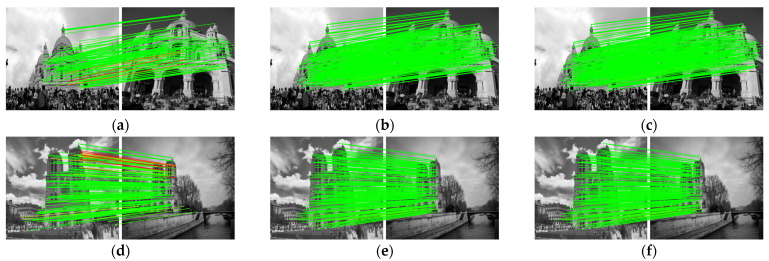
The visualized matching result on YFCC100M. The red lines represent mismatches, and the green lines represent correct matching. (**a**) The results of OANet. (**b**) The results of SuperGlue. (**c**) The results of AdaSG. (**d**) The other results of OANet. (**e**) The other results of SuperGlue. (**f**) The other results of AdaSG.

**Figure 12 sensors-22-05992-f012:**
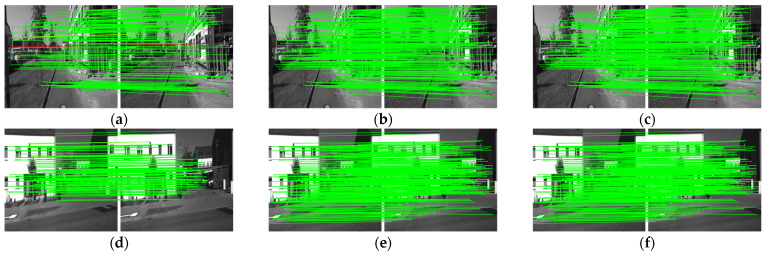
The visualized matching result on KITTI. The red lines represent mismatches, and the green lines represent correct matching. (**a**) The results of OANet. (**b**) The results of SuperGlue. (**c**) The results of AdaSG. (**d**) The other results of OANet. (**e**) The other results of SuperGlue. (**f**) The other results of AdaSG.

**Figure 13 sensors-22-05992-f013:**
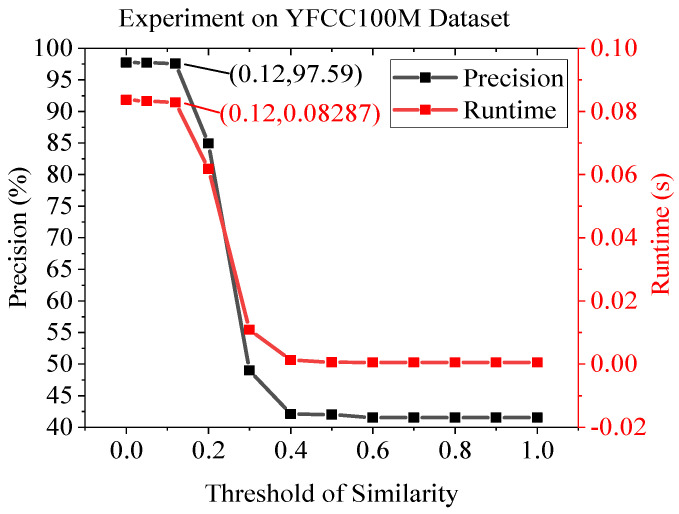
The impact of threshold of similarity on matching precision and average runtime on YFCC100M dataset.

**Figure 14 sensors-22-05992-f014:**
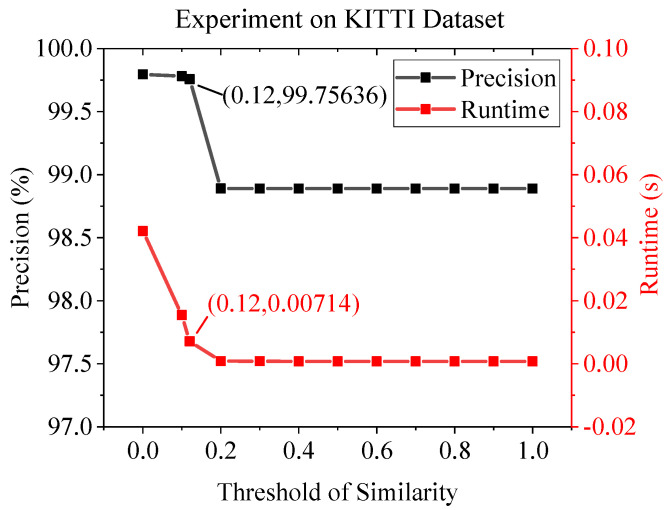
The impact of threshold of similarity on matching precision and average runtime on KITTI dataset.

**Figure 15 sensors-22-05992-f015:**
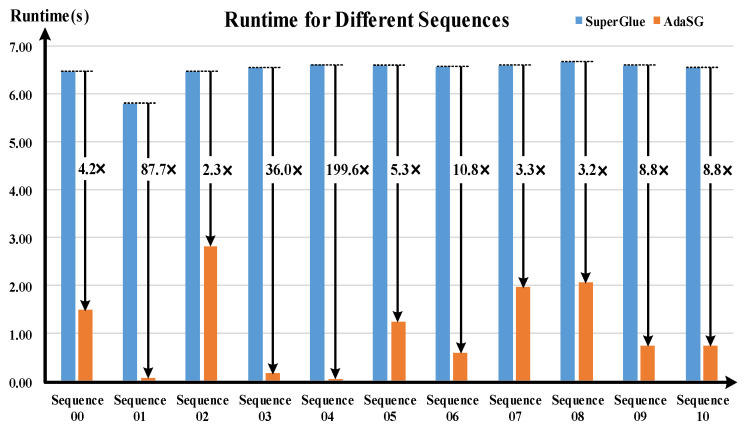
The runtime for different sequences of KITTI.

**Figure 16 sensors-22-05992-f016:**
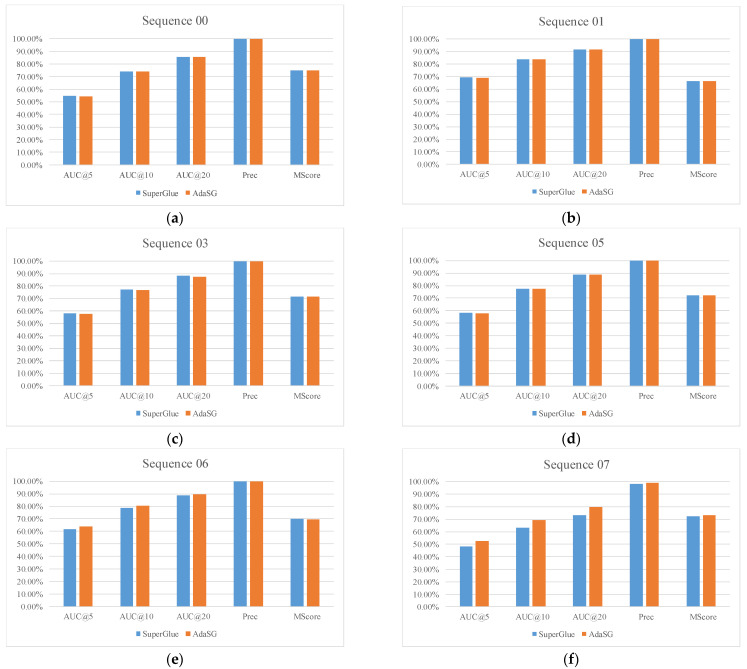
The matching performance for different sequences of KITTI. (**a**–**h**) is the sequence 00–10 of KITTI, respectively.

**Table 1 sensors-22-05992-t001:** Parameter of SuperGlue and AdaSG.

Method	Parameters
SuperGlue	12.1 M
AdaSG (E-mode)	0
AdaSG (D-mode)	12.1 M

**Table 2 sensors-22-05992-t002:** Indoor Experiment on the Scannet dataset.

Matcher	AUC (%)			P (%)	MS (%)	Average Runtime (s)
	@5°	@10°	@20°			
OANet	0.15	0.71	2.35	96.28	44.56	0.016
PointCN	0.28	0.64	2.18	97.38	41.67	0.006
SuperGlue	0.37	1.36	4.82	96.62	76.30	0.043
**AdaSG**	**0.38**	**1.38**	**4.79**	**96.48**	**75.59**	**0.001**

**Table 3 sensors-22-05992-t003:** Outdoor experiment on the YFCC100M dataset.

Matcher	AUC (%)			P (%)	MS (%)	Average Runtime (s)
	@5°	@10°	@20°			
OANet	23.92	41.47	58.65	84.24	15.58	0.021
PointCN	22.04	38.54	55.67	73.13	17.06	0.004
SuperGlue	37.94	58.3	74.59	97.74	23.00	0.092
ClusterGNN	35.31	56.13	73.56	N/A	N/A	N/A
**AdaSG**	**37.90**	**58.23**	**74.52**	**97.64**	**22.91**	**0.088**

**Table 4 sensors-22-05992-t004:** Outdoor Experiment on KITTI Dataset.

Matcher	AUC (%)			P (%)	MS (%)	Average Runtime (s)
	@5°	@10°	@20°			
OANet	49.83	68.32	80.32	99.30	45.88	0.010
PointCN	38.58	68.32	80.32	99.70	38.59	0.003
SuperGlue	61.75	78.60	88.20	99.79	69.09	0.041
**AdaSG**	**61.83**	**78.56**	**88.12**	**99.75**	**69.05**	**0.007**

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
