# Peer review of "AdaSG: A Lightweight Feature Point Matching Method Using Adaptive Descriptor with GNN for VSLAM"

_sensors, 2022, doi:10.3390/s22165992_

Round 1
Reviewer 1 Report
This paper presented a lightweight feature point matching method named AdaSG based on SuperGlue, one of the top image matching methods. Compared with the traditional SuperGlue with GNN, this work's contribution to image matching is making the GNN suitable for resource-constrained devices by adopting an adaptive adjusting architecture according to the similarity of the input image pair to achieve high performance.
Below are several comments:
1. For the Outdoor Experiment on YFCC100M Dataset, the AdaSG performs almost the same as the SuperGlue on Runtime. The reason, which the author claimed, is that the YFCC100M dataset mainly contains non-sequence images. The two photos are visually similar. How to judge their similarity? Would you please describe this part in detail?
2. The quality of Fig.12, Fig.14, and Fig.15 are not good enough. Please improve their quality.
Author Response
Dear Editor and Reviewers,
Thank you very much for your comments and suggestions. They are valuable and helpful for us to improve the quality of our paper. We have carefully gone through all the comments and revised the paper accordingly. We hope that our response below could help address your questions. Please kindly let us know if you have any further questions. Many thanks again for your time!
Regards,
Authors
Comment 1
This paper presented a lightweight feature point matching method named AdaSG based on SuperGlue, one of the top image matching methods. Compared with the traditional SuperGlue with GNN, this work's contribution to image matching is making the GNN suitable for resource-constrained devices by adopting an adaptive adjusting architecture according to the similarity of the input image pair to achieve high performance.
Response:
Many thanks for your time and comments.
Comment 2
For the Outdoor Experiment on YFCC100M Dataset, the AdaSG performs almost the same as the SuperGlue on Runtime. The reason, which the author claimed, is that the YFCC100M dataset mainly contains non-sequence images. The two photos are visually similar. How to judge their similarity? Would you please describe this part in detail?
Response:
Many thanks for your comment. The similarity of the two images is calculated using the sum of absolute differences (SAD), as shown in (1).
Dscore = ∑∑ | I_A(i,j) - I_B(i,j) | (1)
Where Dscore represents the similarity, I_A(i,j) and I_B(i,j) represents the pixel grayscale value of image A and image B at (i,j).
The Dscore will be low when two photos are visually similar, meaning that the viewpoint change or illumination change is small.
Comment 3
The quality of Fig.12, Fig.14, and Fig.15 are not good enough. Please improve their quality.
Response:
Sorry for the poor quality of Fig.12, Fig. 14 and Fig. 15 which is Fig. 16 in the revised manuscript because of layout. As suggested, we have improved their quality in the revised manuscript. (Fig. 12, Fig. 14 and Fig. 16)

Reviewer 2 Report
1.More lightweight methods should be explained in Introduction and Related work.
2.Please explain the different between KPE and FPE module.
3.The authors need to further clarify which are proposed by SuperGlue and which are improved by the authors in the Methods section
4.The authors need to give a comparison of the parameters of different methods.
5.The runtime should be based on the embedded system board and not the Tesla v100 GPU.
6.The authors should add comparisons with recent methods (> 2020)
Author Response
Dear Editor and Reviewers,
Thank you very much for your comments and suggestions. They are valuable and helpful for us to improve the quality of our paper. We have carefully gone through all the comments and revised the paper accordingly. We hope that our response below could help address your questions. Please kindly let us know if you have any further questions. Many thanks again for your time!
Regards,
Authors
Comment 1
More lightweight methods should be explained in Introduction and Related work.
Response:
Many thanks for your time and comments. In the revised manuscript, we have added some lightweight methods in Introduction and Related work. (Section I and Section II)
Comment 2
Please explain the different between KPE and FPE module.
Response:
Many thanks for your time and comments. Sorry for the unclarity about the KPE and the FPE in the original manuscript. In fact, the keypoint encoder (KPE) is also calles featue point encoder (FPE). We have used FPE uniformly and explained the difference between KPE and FPE in the revised manuscript. (Fig. 2)
Comment 3
The authors need to further clarify which are proposed by SuperGlue and which are improved by the authors in the Methods section.
Response:
Many thanks for your time and comments.
SuperGlue proposed an attentional GNN and an optimal matching layer. Based on SuperGlue, We proposed AdaSG, using an adpative architecture to reduce the computational complexity while achieving good matching performance.
In AdaSG, a Similarity Evaluation Module was proposed to judge the similarity of input image pairs and control the mode of AdaSG. Compared to SuperGlue, AdaSG could control the work modes(i.e. D-mode and E-mode) according to similarity of input image pairs. If similarity is low, AdaSG will turn on D-mode and if similarity is high, AdaSG will turn on E-mode.
Adaptive Aggregation Module will activate GNN in D-mode and de-activate GNN in E-mode. What’s more, in D-mode, Outlier Rejection Module will choose inner product to compute the distance of feature points and set the number of sinkhorn iteration large. In E-mode, Outlier Rejection Module will choose Euclidean distance to compute the distance of feature points and set the number of sinkhorn iteration small.
Comment 4
The authors need to give a comparison of the parameters of different methods.
Response:
Many thanks for your time and comments. As suggested, we have added a table to show the comparison of the parameters of AdaSG and SuperGlue in the revised manuscript. (Table 1)
Table 1.Parameter of SuperGlue and AdaSG
method |
parameters |
SuperGlue |
12.1M |
AdaSG(E-mode) |
0 |
AdaSG(D-mode) |
12.1M |
Comment 5
The runtime should be based on the embedded system board and not the Tesla v100 GPU.
Response:
Many thanks for your time and comments.
The runtime based on the Tesla v100 GPU and the embedded system are both shown in the manuscript. The runtime based on the embedded system is shown in Fig15. The SuperGlue and AdaSG are implemented on RK3399Pro, which has four ARM-A53 cores at 1.4 GHz with 4Gb DDR3, for the KITTI dataset. The KITTI contains sequence images, but the moving speed is quite high as the images are taken in a car moving. Therefore, it is a mixture of similar and non-similar image pairs. This causes the D-Mode and E-Mode to be activated in a hybrid fashion which can show the difference between SuperGlue and AdaSG more clearly.
Comment 6
The authors should add comparisons with recent methods (> 2020)
Response:
Many thanks for your time and comments. For fair comparison, we have compared with ClusterGNN[1] on YFCC100M and added the result in the revised paper. (Table 3)
[1] Shi, Yan, et al. "ClusterGNN: Cluster-based Coarse-to-Fine Graph Neural Network for Efficient Feature Matching." Proceedings of the IEEE/CVF Conference on Computer Vision and Pattern Recognition. 2022.

Round 2
Reviewer 2 Report
All my corcerns are solved